# Epidemiology and SARS-CoV-2 Infection Patterns among Youth Followed at a Large Los Angeles Health Network during 2020–2022: Clinical Presentation, Prevalent Strains, and Correlates of Disease Severity

**DOI:** 10.3390/vaccines11061024

**Published:** 2023-05-25

**Authors:** Tawny Saleh, Trevon Fuller, Mary C. Cambou, Eddy R. Segura, Edwin Kamau, Shangxin Yang, Omai B. Garner, Karin Nielsen-Saines

**Affiliations:** 1Department of Pediatrics, Division of Infectious Diseases, David Geffen School of Medicine at UCLA, University of California Los Angeles, Los Angeles, CA 90095, USA; 2Institute for the Environment and Sustainability at UCLA, University of California Los Angeles, Los Angeles, CA 90095, USA; 3Department of Medicine, Division of Infectious Diseases, David Geffen School of Medicine at UCLA, University of California Los Angeles, Los Angeles, CA 90095, USA; 4Facultad de Ciencias de la Salud, Universidad de Huánuco, Huánuco 10260, Peru; esegura@mednet.ucla.edu; 5Department of Clinical Microbiology and Pathology, David Geffen School of Medicine at UCLA, University of California Los Angeles, Los Angeles, CA 90095, USA; edwin.kamau.mil@health.mil (E.K.);

**Keywords:** youth, pediatric COVID-19, VOCs, SARS-CoV-2 disease severity, Los Angeles County, COVID-19 vaccination

## Abstract

Background: Outcomes of SARS CoV-2 infection in infants, children and young adults are reported less frequently than in older populations. The evolution of SARS-CoV-2 cases in LA County youths followed at a large health network in southern California over two years was evaluated. Methods: A prospective cohort study of patients aged 0–24 years diagnosed with COVID-19 was conducted. Demographics, age distribution, disease severity, circulating variants of concern (VOCs), and immunization rates were compared between first and second pandemic years. Logistic regression estimated odds ratios (OR) and 95% confidence intervals (CI) of factors associated with severe/critical COVID-19. Results: In total, 61,208 patients 0–24 years of age were tested for SARS-CoV-2 by polymerase chain reaction (PCR); 5263 positive patients (8.6%) with available data were identified between March 2020 and March 2022. In Year 1, 5.8% (1622/28,088) of youths tested positive, compared to 11% (3641/33,120) in Year 2 (*p* < 0.001). Most youths had mild/asymptomatic illness over two years. SARS-CoV-2 positivity was >12% across all age groups in the second half of Year 2, when Omicron prevailed. Pulmonary disease was associated with higher risk of severe COVID-19 in both years (OR: 2.4, 95% CI: 1.4–4.3, *p* = 0.002, Year 1; OR: 11.3, 95% CI: 4.3–29.6, Year 2, *p* < 0.001). Receipt of at least one COVID-19 vaccine dose was protective against severe COVID-19 (OR: 0.3, 95% CI: 0.11–0.80, *p* < 0.05). Conclusions: Despite different VOCs and higher rates of test positivity in Year 2 compared to Year 1, most youths with COVID-19 had asymptomatic/mild disease. Underlying pulmonary conditions increased the risk of severe COVID-19, while vaccination was highly protective against severe disease in youths.

## 1. Introduction

By 30 March 2023, 26,415,622 youths had COVID-19 in the United States, representing 32% of reported cases [1]. This figure is likely an underestimate given the widespread use of home-based SARS-CoV-2 antigen testing with results unreported and the typical asymptomatic/mild disease presentation in most youths, leading to under-reported infections [2,3]. Asymptomatic youths have epidemiological importance as they transmit SARS-CoV-2 to others as their nasopharyngeal viral loads are comparable or higher than those of adults [4,5,6,7,8]. During Year 2 of the COVID-19 pandemic (March 2021–March 2022), we saw the serial emergence of highly transmissible variants of concern (VOCs) and concurrent distribution of effective vaccines to the general population, including youths 5 years and older. The Delta variant emerged in the spring of 2021, rapidly replacing other variants and achieving global dominance by the summer of 2021. Omicron variants were subsequently identified, rapidly spreading across Europe and North America by late December 2021, peaking by January 2022 [9].

By February 2022, 75% of U.S. youths had serologic evidence of SARS-CoV-2 infection, with approximately 33% becoming newly seropositive since December 2021 [2]. The greatest increase in seroprevalence from September 2021 to February 2022 occurred in younger age groups with the lowest vaccination coverage. The proportion of the U.S. population fully vaccinated by April 2022 increased with age (5–11: 28%; 12–17: 59%; 18–49: 69%) [2]. Infection with highly contagious Omicron variants did not appear to protect against reinfection and long-term morbidities, including long COVID [2]. Although most youths with COVID-19 exhibited mild symptoms, a rise in hospitalizations was noted during the time of circulation of both Delta and Omicron variants [10]. The proportion of admitted youths requiring intensive care was similar during the Delta surge but lower when Omicron variants predominated, compared to earlier circulating strains [10,11]. COVID-19-related deaths in youths remained rare, with greater numbers reported among children less than 12 months and adolescents aged 15–19 years [12].

The purpose of this study was to characterize the evolution of COVID-19 in Los Angeles (LA) County youths followed at our institution, comparing clinical patterns during the first two pandemic years. There is an urgent need to collect more information on the immediate clinical repercussions of COVID-19 in youths and how disease evolved over time. A better understanding of disease presentation in youths has important implications, given the dynamic nature of SARS-CoV-2, its association with complications such as Multisystem Inflammatory Syndrome in Children (MIS-C), and the protective role of COVID-19 vaccines. More information helps guide policy, vaccine development and emerging treatments for management of short- and long-term complications in youths.

## 2. Methods

During the first two years of the COVID-19 pandemic, we conducted a longitudinal study evaluating a youth cohort 0 to 24 years of age diagnosed with SARS-CoV-2 in Los Angeles (LA) County followed at a large health network. The first year encompassed patients diagnosed between mid-March 2020 and March 2021 [13]. All youths were tested by reverse transcriptase polymerase chain reaction (RT-PCR) for SARS-CoV-2 at our institution. Reinfection was defined as a positive RT-PCR test ≥90 days after the first infection. Research methods have been previously described [14]. Our setting comprised a large academic medical center (Ronald Reagan Medical Center and Mattel Children’s Hospital at UCLA), a community hospital (UCLA Santa Monica Hospital), and a widespread network of UCLA-affiliated clinics throughout all of LA County. All participants were tested for SARS-CoV-2 by RT-PCR in a single laboratory. Confirmed cases of COVID-19 were defined as positive RT-PCR results. If a participant had multiple tests performed, the first positive result was used in the analysis. Each participant was counted once, even if more than one RT-PCR test was performed. Youths with COVID-19 in pandemic Year 1 from March 2020 to 2021 [13,14] were compared with youths enrolled between 1 April 2021 to 31 March 2022 (pandemic Year 2), coinciding with a heightened COVID-19 surge in LA County in January 2022. Data abstracted from electronic medical records included demographics, time of infection, clinical findings, immunization status, and disease severity. All youths were classified as having asymptomatic, mild, moderate, severe, or critical COVID-19 according to the NIH Coronavirus Disease Treatment Guidelines [15]. Data on youths admitted with a clinical diagnosis of MISC-C [16,17] were collected over the same period with findings described. Study activities were approved by the UCLA Institutional Review Board which provided an IRB exemption.

All SARS-CoV-2 RT-PCR tests were performed by the UCLA Clinical Microbiology Laboratory as previously described [14]. Using genomic data on SARS-CoV-2 from the Global Initiative on Sharing Avian Influenza Data (GISAID) initiative [9], we calculated the frequency of SARS-CoV-2 VOCs and common lineages by month from March 2020 to March 2022.

We performed a descriptive analysis of demographics of all patients aged less than 25 years with positive SARS-CoV-2 RT-PCR results in our health system. We performed a chi-square test to compare basic demographic information among youths detected for SARS-CoV-2 at our institution during the two pandemic years. Disease severity was transformed into categorical data. Individuals with no clinical data or individuals tested outside of our institution were excluded from further analysis.

The primary outcome of the study was severe/critical COVID-19. To analyze the outcome, participants were divided into those recruited in the first and second years of the pandemic. In each group, we examined the association between outcome and the following predictor variables: sex, age (<1, 1–5, 6–11, 12–18, and 19–24 years), race/ethnicity, insurance coverage, comorbidities, occupation, and receipt of one or more doses of the COVID-19 vaccine. Vaccination was analyzed in the second year as it was not widely available in the first year. Logistic regression analyses used SPSS v.19 (IBM, Inc., Armok, NY, USA) to estimate odds ratios (OR) and 95% confidence intervals (CI) for factors associated with severe/critical disease.

## 3. Results

Over two years, a total of 92,673 RT-PCR tests for SARS-CoV-2 were performed on 61,208 youths at our institution. In Year 1, 36,239 RT-PCR tests were performed in 28,088 individuals, with 1847 of 28,088 individuals testing positive (6.6%) [13,14]. In the second year, 56,434 SARS-CoV-2 RT-PCR assays were performed in 33,120 youths, with 3824 youths (11.5%) being positive for SARS-CoV-2. (Figure 1). The frequency and number of COVID-19 cases in our study was significantly higher in Year 2 as compared to Year 1 of the pandemic (Table 1), matching LA County’s surveillance data for the same period [18]. In Years 1 and 2, respectively, 225 and 183 youths with positive results had no clinical information available and were excluded from further analysis. Data were analyzed for 5263 individuals: 1622 in Year 1 and 3641 in Year 2 (Table 1).

Over time, (Table 1), there was an age shift among confirmed COVID-19 cases from older youths (19–24 years) to younger age groups, with a mean age of 14.3 years in the first year and 11.4 years in the second year, *p* < 0.001 (Table 1). As time progressed, younger youths were identified with COVID-19. This was most notable among children less than 5 years, especially among toddlers and infants. Infection frequency increased in this age group from 19.4% in Year 1 to 31.1% in Year 2. Seven youths died: three in Year 1 (0.2%) and four in Year 2 (0.1%); all had underlying medical conditions and severe COVID-19.

Most youths positive for SARS-CoV-2 had mild/asymptomatic illness, regardless of study period (Table 1). When both Year 1 and Year 2 were compared; youths testing positive in the second year tended to have more asymptomatic/mild/moderate illness than those in Year 1, (99% vs. 94%, respectively, *p* < 0.001, Table 1). In total, 133 youths had severe/critical illness: 98 (6%) in Year 1 and 35 in Year 2 (1%). A statistically significant difference in disease severity across all age groups was noted between Years 1 and 2 (Table 2). Disease severity declined in children less than 12 months, although this change was not statistically significant. Significant *p* values < 0.05 are in bold. 

Race/ethnicity was available for 5149 youths over two years. Compared with other ethnic groups, Hispanic youths continued to make up the largest ethnic group with positive SARS-CoV-2 results (i.e., approximately 50% of the LA County population is Hispanic). However, there was a significant rise in the second year of the pandemic in White and Asian youths diagnosed with COVID-19, from 29.8% to 37.8% and 3.4% to 9.3%, respectively, *p* < 0.001 (Table 1). During the first year, a higher percentage of older patients employed as front-line workers (10.6%) tested positive, as compared to the second year (4.1%) where nearly all positive youths in this age group were students. A higher proportion of patients with private/health maintenance organization insurance were identified in the second year as compared to the first year (80.1% versus 70.3%, Table 1, *p* < 0.001). Conversely, the first pandemic year had a higher proportion of patients with Medical/Safety Net insurance as compared to the second year; 27.1% versus 20%, *p* < 0.001 (Table 1). Significant *p* values < 0.05 are in bold.

From October 2020 to March 2022, several SARS-CoV-2 VOCs emerged (Figure 2). In the second half of first year, the most common strain circulating in LA County was the Epsilon variant (B.1.247, B.1.249). As the surge continued, other predominant variants, including the Alpha variant (B.1.1.7), started emerging. By the end of the first pandemic year, the Delta variant (B.1.617 and derivative lineages) began to emerge, dominating the first half of the second year. By the end of the second year, the highly transmissible Omicron (B.1.1.529) emerged [9], causing the highest frequency of infections in all age groups, followed by the Epsilon variant (Figure 3).

Vaccination against SARS-CoV-2 became widely available in Year 2, with high rates of receipt of at least one vaccine dose in older youths (94.4% in 19–24-year-olds, 84.8% in 12–18-year-olds, Table 2). In contrast, receipt of at least one vaccine dose was significantly less in younger age groups, as the FDA approval of vaccines for children under 12 years did not come through until the end of the study period. Immunization of children under five years was not recommended during the study period.

Risk factors for COVID-19 disease severity are shown in Figure 4 and Appendix A. Pulmonary disease was associated with a high risk of severe COVID-19 in both years (OR: 2.4, 95% CI: 1.4–4.3, *p* = 0.002 in Year 1, and OR: 11.3, 95% CI: 4.3–29.6 in Year 2, *p* < 0.001 Figure 4, Appendix A). Cardiac disease was associated with severe COVID-19 risk in Year 2 (OR: 4.3, 95% CI: 1.5–12.2, *p* = 0.006). In Year 1, the risk of severe/critical COVID-19 was significantly lower in children 1 to 5 years of age (OR: 0.3, 95% CI: 0.15–1.82, *p* = 0.012) and 6 to 11 years of age (OR: 0.41, 95% CI: 0.11–0.77, *p* = 0.043) than in adolescents and youths 19–24 years of age (Figure 4). In Year 2, age was not associated with disease severity. Sex, race/ethnicity, and insurance coverage were not associated with disease severity (Figure 4, Appendix A). The receipt of at least one dose of the COVID-19 vaccine was a protective risk factor against disease severity in the overall cohort (OR: 0.3, 95% CI 0.11–0.80, *p* < 0.15) once vaccines became available (Figure 4, Appendix A).

Pulmonary comorbidities increased the risk of severe/critical COVID-19 in both years. In Year 1, younger youths had a lower risk of severe/critical outcomes [13,14]. In Year 2, in addition to pulmonary comorbidities, cardiac comorbidities increased the risk of severe/critical COVID-19.

During the two years, a total of 126 repeat SARS-CoV-2 infections in 5265 youths (2.4%) were recorded. The average duration between repeat infections was 246 days. Among patients with repeat COVID-19, 49% received at least one dose of SARS-CoV-2 vaccine; 123 patients with re-infection (97.6%) had mild/asymptomatic illness. Three youths with severe disease upon re-infection were unvaccinated and had chronic diseases requiring immunosuppression. Appendix A describes the demographic characteristics of youth with reinfection.

In Year 2, 54 of 3641 youths (1.5%) had concurrent viral co-infections in combination with SARS-CoV-2. The great majority of co-infections were due to RSV (n = 44, 81.5%). Three patients with SARS-CoV-2 and RSV had severe manifestations requiring oxygen and a higher level of care. Two patients had chronic medical conditions while one had no pre-existing conditions. Other viruses co-infecting youths with SARS-CoV-2 infection in Year 2 included influenza A (n = 5), influenza B (n = 1), rhino/enterovirus (n = 3), and adenovirus (n = 1). These co-infections were mild and occurred in healthy children.

Twenty-five cases of MIS-C were identified in two years, resulting in two deaths (8% mortality). Eleven cases occurred in the first year (average age 12.5 years) [13,14]. Three children had chronic medical conditions, while the remaining were healthy. One patient with pre-existing immunosuppression died after a complicated hospital course. In the second year, 14 patients developed MIS-C (average age 6.4 years). Thirteen children had no pre-existing health conditions while one child with an underlying medical disorder expired from MIS-C complications. Only 3 of 14 youths (21.4%) in Year 2 received two doses of COVID-19 vaccine prior to developing MIS-C; the others were unvaccinated.

## 4. Discussion

Los Angeles County is one of the largest and most socioeconomically diverse counties in the United States, with over 10 million residents; one-third of the population is younger than 25 years of age [19]. Early in the pandemic, a predominance of older, front-line working youths with COVID-19 was noted, with the majority of participants being Hispanic [14]. Over time, the pandemic shifted to include more individuals of other ethnic backgrounds, and a broader representation of all individuals within the socioeconomic strata, including younger children. From Years 1 to 2, infection also shifted from the more medically vulnerable youth population to a healthier group of children, adolescents, and young adults.

Although assessing the effect of age on SARS-CoV-2 transmission was beyond the scope of our study, a systematic review and meta-analysis of household transmission early in the pandemic reported that the secondary attack rate was higher for adult index cases than for pediatric ones [20]. It has been hypothesized that prolonged infection in the immunosuppressed gives rise to the emergence of new SARS-CoV-2 variants. A number of case studies of such infection have analyzed individuals sixty years of age and older [21,22]. However, the available data are insufficient to attribute the emergence of variants to a particular age group.

We witnessed a rampant spread of new VOCs among all socioeconomic classes during the second year, while in the first year the virus predominantly impacted urban, underserved communities [13,14]. Youths in the first half of Year 1 had a near 10-fold higher proportion of severe disease than youths infected in the second half of Year 2: 17% vs. 1.8%, respectively [13,14]. Age increase was associated with greater disease severity during the first pandemic year [13].

A considerably higher number of youths were infected in the second pandemic year, with a much higher frequency of infection among tested individuals. A higher infection rate was noted in females, non-Hispanic ethnic groups, patients with private insurance, students, and healthy individuals. As the pandemic progressed, younger age groups were more frequently affected, with infection prevalence rising 67%. From Years 1 to 2, the pandemic shifted from a more socioeconomically and medically vulnerable youth population to a less disenfranchised, healthier group of children, adolescents and young adults.

Disease severity dropped from 6% to 1% from the first to second year, across all age groups. Although the frequency of COVID-related deaths was minimal, it also halved in the second year. As such, the proportion of severe/critical cases diminished significantly over time, likely owing to the age shift to younger age groups, the circulation of less virulent VOCs, and the development of immunity from widespread vaccination and likely natural infection, especially in older youths. As in previous studies, our youth tended to be asymptomatic or have mild symptoms [10,23,24,25,26,27,28,29]. In a cohort study of Canadian youths, Delta and Omicron variants were more strongly associated with less severe symptoms when compared to original virus types, as in our study. The study noted no differences in undesirable outcomes (hospitalization and intensive care unit admission) across VOCs [30]. In contrast, we noted a significant improvement in outcomes over time.

Repeat SARS-CoV-2 infections were identified in a small percentage of youths (2.4%), being mild to asymptomatic in 98%. Three unvaccinated patients with co-morbidities had severe disease upon re-infection. No viral co-infections were noted in the first year of the pandemic, but in the second, 1.5% of our population had concurrent viral co-infections, most notably RSV; only 6% had more severe disease. Concurrent viral infections during COVID-19 have been similarly reported in children in other countries, with most having mild symptoms [31,32]. Interestingly, our results are consistent with mathematical modeling studies evaluating the role of viral interference in RSV and SARS-CoV-2 concurrent infections, suggesting that some viruses have the ability to suppress the growth of other viruses when simultaneously present in the same host [33]. RSV may suppress infection with SARS-CoV-2 by triggering the production of interferons that stimulate uninfected cells to adopt a protective state [34].

MIS-C admissions were rare at our institution, with only 25 cases recorded over two years. Many of these patients did not have positive PCR results at our institution before developing MIS-C. Nevertheless, despite being a rare event, MIS-C was associated with a high mortality of 8% in patients with pre-existing conditions. The majority of MIS-C cases occurred in unvaccinated patients. Studies have shown that the clinical course of MIS-C has been consistent across different pandemic waves, although incidence seems to be decreasing, potentially due to widespread immunization efforts [35].

Due to hypoxia, coagulopathy, and inflammation from infection, COVID-19 has been associated with myocarditis, pulmonary hemorrhage and embolisms, and brain ischemia and hemorrhage [36,37,38]. Consequently, COVID-19 may adversely impact multiple organ systems as both an acute infection or post-infectious inflammatory disease process, such as MIS-C, leading to significant morbidity and mortality. Although most youths have self-limited disease, severe COVID-19 complications can occur in this age group. This underscores the importance of COVID-19 vaccination for individuals of all ages.

### Limitations

There were several strengths to our study, including a robust sample size of socioeconomically and ethnically diverse youth with RT-PCR-confirmed SARS-CoV-2 infection. Patients were identified during different periods of the pandemic, yielding sufficient data to characterize COVID-19 outcomes over an extended period. One limitation is that we were unable to include youths diagnosed by rapid antigen testing, a practice that became increasingly common in the second pandemic year. This likely led to an underestimation of cases, particularly in Year 2. While less accurate than nucleic acid tests, rapid antigen tests appear to be best suited for identifying SARS-CoV-2 infection in individuals who are symptomatic in high prevalence settings [39,40]. Perhaps we could have enrolled additional participants during the peak of the Delta or Omicron waves if we had used antigen testing to screen symptomatic youths. Another limitation is that testing patterns shifted over time, including access to RT-PCR testing, potentially impacting case identification and favoring the detection of a higher number of asymptomatic/mild cases in later periods as testing accessibility increased. This pattern, however, would unlikely affect severe/critical cases as those sought medical attention.

## 5. Conclusions

In summary, our study characterized the evolution of SARS-CoV-2 infection in L.A County youths over two years within a widespread health system. We noted a shift in the pandemic profile from vulnerable, disenfranchised older patients who tended to have more severe disease to a more universal, younger age group representative of all segments of socioeconomic strata, including a shift to more females identified with COVID-19 in the second year. A large proportion of youths 6 years and older received vaccines in the second year, and vaccination was protective against severe disease. Pulmonary and cardiac co-morbidities predicted severe disease throughout the two years. MIS-C admissions were rare but carried a high mortality in patients with co-morbidities. The ongoing monitoring of COVID-19 presentation in children, adolescents and young adults will determine if outcomes continue to improve over time following wider vaccine availability.

## Figures and Tables

**Figure 1 vaccines-11-01024-f001:**
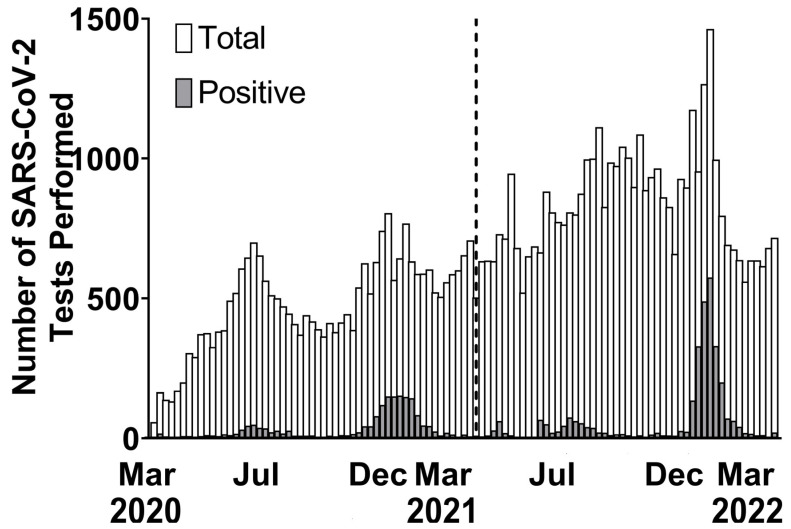
Youth SARS-CoV-2 testing by week, March 2020–March 2022.

**Figure 2 vaccines-11-01024-f002:**
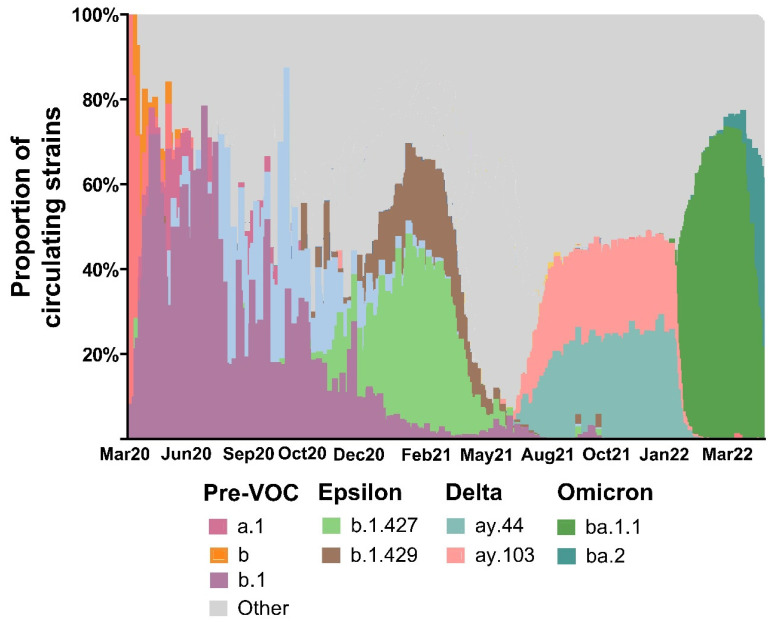
Circulating variants of concern (VOC) in California by month and year.

**Figure 3 vaccines-11-01024-f003:**
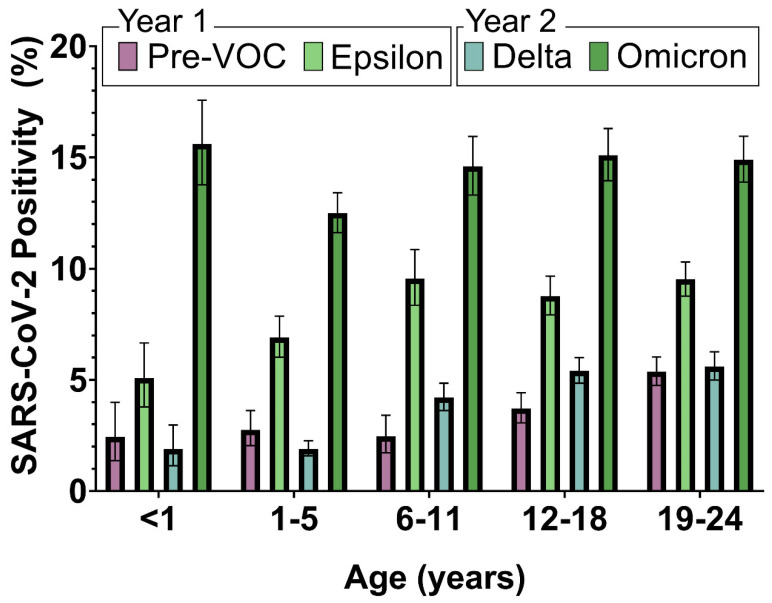
Percent of positive SARS-CoV-2 tests by age.

**Figure 4 vaccines-11-01024-f004:**
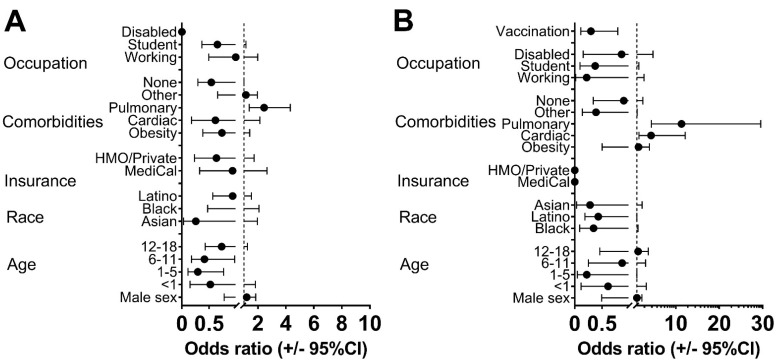
Each odds ratio represents the risk of severe/critical COVID-19 associated with the variable listed on the *y*-axis compared to a reference class. The reference classes were female for sex, 19–24 for age, White for race, uninsured for insurance coverage, unemployed for working, not in school for student, not disabled for disability status, and unvaccinated for vaccination status. (**A**) represents Year 1, and (**B**) represents Year 2.

**Table 1 vaccines-11-01024-t001:** Demographics, comorbidities, and disease severity in Los Angeles County youth.

	13 March 2020–31 March 2021 N = 1622		1 April 2021–31 March 2022 N = 3641		*p*
Sex	N	%	N	%	0.009
Female	795	49%	1927	52.9%	
Male	827	51%	1714	47.1%	
**Age group at testing**					**<0.001**
<1 year	59	3.6%	248	6.8%	
1–5 years	257	15.8%	886	24.3%	
6–11 years	209	12.9%	672	18.5%	
12–18 years	468	28.8%	959	26.3%	
19–24 years	630	38.8%	876	24.1%	
Mean Age (SD)	14.3	7.5	11.4	7.7	**<0.001**
**Race/ethnicity**					**<0.001**
Asian	54	3.4%	337	9.3%	
Black/African American	172	10.8%	409	11.3%	
Hispanic/Latino	839	52.6%	1477	40.7%	
White	475	29.8%	1374	37.8%	
Other	55	3.4%	35	1%	
Missing	27		9		
**Type of Insurance**					**<0.001**
Uninsured	55	3.4%	215	5.9%	
Medical/Safety Net	384	23.7%	510	14%	
HMO/Private	1184	73%	2916	80.1%	
**Comorbidities**					
Obesity	196	12.1%	344	9.4%	**0.004**
Pulmonary	173	10.7%	415	11.4%	0.432
Cardiac	56	3.5%	70	1.9%	**0.01**
Other	441	27.2%	245	6.7%	**<0.001**
None	1088	67%	2792	76.7%	**<0.001**
**Occupation (Patient)**					
Working	172	10.6%	149	4.1%	**<0.001**
Student	1216	74.9%	3216	88.3%	**<0.001**
Disabled	28	1.7%	44	1.2%	0.136
Other	103	6.3%	240	6.6%	0.739
**Disease Severity**					**<0.001**
Asymptomatic	394	24%	706	19%	
Mild/moderate	1127	70%	2900	80%	
Severe or Critical	98	6%	35	1%	
Death	3	0.18%	4	0.11%	0.45

**Table 2 vaccines-11-01024-t002:** Severe/critical COVID-19 and COVID-19 vaccination rates by age group in LA County youth.

		Severe/Critical			Vaccination Rate		
		Year 1	Year 2	*p*	Year 1	Year 2	*p*
**Age (years)**	<1, N (%)	3/59 (5.1%)	4/248 (1.6%)	0.12	0/59 (0%)	0/248 (0%)	0.99
	1–5, N (%)	5/257 (1.9%)	5/886 (0.6%)	**<0.001**	0/257 (0%)	83/886 (9.4%)	**<0.001**
	6–11, N (%)	7/209 (3.3%)	7/672 (1.1%)	**0.03**	0/209 (0%)	452/672 (67.3%)	**<0.001**
	12–18, N (%)	27/468 (5.8%)	10/959 (1.1%)	**<0.001**	21/468 (4.5%)	813/959 (84.8%)	**<0.001**
	19–24, N (%)	56/630 (8.9%)	9/876 (1%)	**<0.001**	15/630 (2.4%)	827/876 (94.4%)	**<0.001**

## Data Availability

De-identified data is available upon reasonable request.

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
