# Peer review of "Epidemiology and SARS-CoV-2 Infection Patterns among Youth Followed at a Large Los Angeles Health Network during 2020–2022: Clinical Presentation, Prevalent Strains, and Correlates of Disease Severity"

_vaccines, 2023, doi:10.3390/vaccines11061024_

Round 1
Reviewer 1 Report
The authors present an interesting prospective longitudinal study on the response to COVID 19 prevalence.
it would be interested if the authors include in their study which age range represented the privileged reservoir of contagion and if variants are caused by spreading of virus among young people.
Please, please format the figure 4, I cannot read it.
In the discussion, it is demanded to underline the importance of the prevention of COVID19 contagion among young people, because of the post-covid multi-organic disease. In fact, SARS-Cov 2 during the infection phase can affect lungs, brain, heart etc. In this case, you could cite these and other works about this topic:
https://doi.org/10.1007/s12024-020-00310-8
https://doi.org/10.1111/bpa.13013
https://doi.org/10.3390/diagnostics11091647
Author Response
The authors present an interesting prospective longitudinal study on the response to COVID 19 prevalence.
Thanks for your assessment of our manuscript.
it would be interested if the authors include in their study which age range represented the privileged reservoir of contagion and if variants are caused by spreading of virus among young people.
We agree that it is important to investigate the influence of age upon SARS-CoV-2 transmission and the emergence of COVID-19 variants. In the Discussion, we added: “Although assessing the effect of age on SARS-CoV-2 transmission was beyond the scope of our study, a systematic review and meta-analysis of household transmission early in the pandemic reported that the secondary attack rate was higher for adult index cases than for pediatric ones [1]. It has been hypothesized that prolonged infection in the immunosuppressed gives rise to the emergence of new SARS-CoV-2 variants. A number of case studies of such infection have analyzed individuals sixty years of age and older [2,3]. However, the available data is insufficient to attribute to the emergence of variants to a particular age group.”
Please, please format the figure 4, I cannot read it.
We apologize. We have reformatted the figure.
In the discussion, it is demanded to underline the importance of the prevention of COVID19 contagion among young people, because of the post-covid multi-organic disease. In fact, SARS-Cov 2 during the infection phase can affect lungs, brain, heart etc. In this case, you could cite these and other works about this topic:
https://doi.org/10.1007/s12024-020-00310-8
https://doi.org/10.1111/bpa.13013
https://doi.org/10.3390/diagnostics11091647
In the Discussion, we added: “Due to hypoxia, coagulopathy, and inflammation from infection, COVID-19 has been associated with myocarditis, pulmonary hemorrhage and embolisms, and brain ischemia and hemorrhage [4-6]. Consequently, COVID-19 may adversely impact multiple organ systems as both an acute infection or post infectious inflammatory disease process, such as MIS-C, leading to significant morbidity and mortality Although most youth have self-limited disease, severe COVID-19 complications can occur in this age group. This underscores the importance of COVID-19 vaccination for individuals of all ages.”

Reviewer 2 Report
Saleh et. al describe in their article the epidemiology and SARS-CoV-2 Infection Patterns among youth between 2020-2022 in the greater LA area.
The paper is well written and the conclusions made are scientifically sound. The limitations of theirs study are clearly described. I have some comments to the authors, how to improve their manuscript.
General comments:
This work is a continuation of their research, already published. Please clearly reference to your previous study in the results section and contrast the difference between the two periods under surveillance
Please describe in the methods section, if re-infection in your study group occurred, and if, how you exactly counted these cases. It might be interesting to include the data on re-infected individuals as a separate table, how frequent this was observed in each subgroup.
Specific points:
Table 1: Can the authors exactly mention the numbers for asymptomatic, mild and moderate cases for each group respectively?
Figure 3 is not mentioned and discussed in the results section. Moreover, it would be clearer, if there would be 3 subgraphs, one for the 2 year period, one for year 1 and one for year 2
Figure 4: Please make A and B similar, so that the same category is written in the same line for a direct comparison. Moreover, write for sex the other real category. Currently the reference is female, so I suppose the risk is for “male” or “not female”? Please clarify.
Line 260 onwards: Can you speculate, why RSV might suppress SARS-CoV-2 infection?
Line 265: MIS-C cases: To better appreciate your results, is it maybe possible that those cases are listed with their characteristics (same as in table 1) in the supplement? Maybe you can further speculate about additional risk factors for this complicated course.
Author Response
Saleh et. al describe in their article the epidemiology and SARS-CoV-2 Infection Patterns among youth between 2020-2022 in the greater LA area.
The paper is well written and the conclusions made are scientifically sound. The limitations of theirs study are clearly described. I have some comments to the authors, how to improve their manuscript.
Thank you. We have responded to each comment in turn below.
General comments:
This work is a continuation of their research, already published. Please clearly reference to your previous study in the results section and contrast the difference between the two periods under surveillance.
In the Methods, we added citations of our previous studies. In the Discussion, we added: “From years 1 to 2, infection also shifted from a more medically vulnerable youth population to a healthier group of children, adolescents and young adults”.
Please describe in the methods section, if re-infection in your study group occurred, and if, how you exactly counted these cases. It might be interesting to include the data on re-infected individuals as a separate table, how frequent this was observed in each subgroup.
In the Methods, we added: “Reinfection was defined as a positive RT-PCR test ³ 90 days after the first infection”. In addition, we added Supplemental Table 2 listing the data on reinfected individuals by subgroup.
Specific points:
Table 1: Can the authors exactly mention the numbers for asymptomatic, mild and moderate cases for each group respectively?
Our database did not differentiate mild and moderate in year 1. For this reason, we only added the numbers of asymptomatic and mild/moderate for each year.
Figure 3 is not mentioned and discussed in the results section. Moreover, it would be clearer, if there would be 3 subgraphs, one for the 2 year period, one for year 1 and one for year 2
Figure 3 is a clustered bar chart in which each cluster of bars represents an age group. Within each cluster there are four bars that stand for the pre-VOC period, the Epsilon variant, the Delta variant, and the Omicron variant. The pre-VOC and Epsilon waves occurred during year 1 and the Delta and Omicron waves during year 2. For this reason, the figure in its current format already shows data for year 1 and year 2. If we were to make a subgraph for year 1, a subgraph for year 2, and a subgraph for the two year period, the year 1 and year 2 subgraphs would repeat the same data as the two year period subgraph. In light of this, we opted not to plot three subgraphs. However, we have added labels and boxes to the legend to explain that pre-VOC and Epsilon correspond to year 1 and Delta and Omicron correspond to year 2.
Figure 4: Please make A and B similar, so that the same category is written in the same line for a direct comparison. Moreover, write for sex the other real category. Currently the reference is female, so I suppose the risk is for “male” or “not female”? Please clarify.
We revised the figure to make A and B similar. The reference class for sex was female, therefore the odds ratio represents male. We added the label “Male sex” to clarify this.
Line 260 onwards: Can you speculate, why RSV might suppress SARS-CoV-2 infection?
We added: “RSV may suppress infection with SARS-CoV-2 by triggering the production of interferons that stimulate uninfected cells to adopt a protective state [7].“
Line 265: MIS-C cases: To better appreciate your results, is it maybe possible that those cases are listed with their characteristics (same as in table 1) in the supplement? Maybe you can further speculate about additional risk factors for this complicated course.
We added a supplemental table with the characteristics of MIS-C cases (Table S1).
Reviewer 3 Report
The authors have followed the outcome of SARS-CoV-2 infections in the age groups newborn to 24 years in Los Angeles County, one of the largest in the United States. The study reported PCR-data and covered the two year period between 3/2020 and 3/2022. The analysis included 61,208 individuals and was conducted by a group of experienced pediatricians and infectious disease researchers. The study period witnessed the emergence of numerous previously unknown variants of SARS-CoV-2.
The major findings are the following. Most PCR-proven SARS-CoV-2-infected probands exhibited no symptoms or mild illness. In Year 1 of the investigation, 5.8%, in year 2, 11% of probands tested PCR positive for SARS-CoV-2. Co-morbidity with pulmonary or cardiac ailments heightened the risk of severe viral infections. At least one COVID-19 vaccine dose protected against severe symptoms in the young (age 0 to 24 years) population. Patients suffering from Multisystem Inflammatory Syndrome in Children were rare in the authors’ study cohort but had a high mortality rate, particularly when suffering from additional co-morbidities.
The manuscript describes a careful and well controlled investigation. Its results will be of general interest to pediatricians, medical microbiologist and virologists. There did not seem to be unexpected results, fundamental aspects of virology were not studied, hence the overall importance of this study remains somewhat limited. Here are some comments on formal aspects of the presentation that could easily be addressed.
(i) The term “protective risk factor” for protection by vaccination seems linguistically odd.
(ii) Figure 3 is certainly OK, but since you have used color in other figures, here color would help the reader.
(iii) In Figure 4: Why does “sex” stand for “female”?
(iv) Page 10: Is “rapid antigen testing” –even under favorable conditions – a reliable test at all?
(v) Line 293 , …. rare, instead of rate.
Author Response
The authors have followed the outcome of SARS-CoV-2 infections in the age groups newborn to 24 years in Los Angeles County, one of the largest in the United States. The study reported PCR-data and covered the two year period between 3/2020 and 3/2022. The analysis included 61,208 individuals and was conducted by a group of experienced pediatricians and infectious disease researchers. The study period witnessed the emergence of numerous previously unknown variants of SARS-CoV-2.
The major findings are the following. Most PCR-proven SARS-CoV-2-infected probands exhibited no symptoms or mild illness. In Year 1 of the investigation, 5.8%, in year 2, 11% of probands tested PCR positive for SARS-CoV-2. Co-morbidity with pulmonary or cardiac ailments heightened the risk of severe viral infections. At least one COVID-19 vaccine dose protected against severe symptoms in the young (age 0 to 24 years) population. Patients suffering from Multisystem Inflammatory Syndrome in Children were rare in the authors’ study cohort but had a high mortality rate, particularly when suffering from additional co-morbidities.
The manuscript describes a careful and well controlled investigation. Its results will be of general interest to pediatricians, medical microbiologist and virologists. There did not seem to be unexpected results, fundamental aspects of virology were not studied, hence the overall importance of this study remains somewhat limited. Here are some comments on formal aspects of the presentation that could easily be addressed.
(i) The term “protective risk factor” for protection by vaccination seems linguistically odd.
We reworded this as: “In year one, the risk of severe/critical COVID-19 was significantly lower in children 1 to 5 years of age (OR: 0.3, 95% CI: 0.15 – 1.82, p=0.012) and 6 to 11 years of age (OR: 0.41, 95% CI: 0.11- 0.77, p = 0.043) than in adolescents and youth 19-24 years of age (Figure 4).”
(ii) Figure 3 is certainly OK, but since you have used color in other figures, here color would help the reader.
We re-did the Figure 3 in color.
(iii) In Figure 4: Why does “sex” stand for “female”?
In the Figure 4 legend, we added “Each odds ratio represents the risk of severe/critical COVID-19 associated with the variable listed on the y-axis compared to a reference class.” For example, the variable “Male sex” in Panel A has an odds ratio of 1.195 (95% CI: 0.775-1.841), which denotes that males have a risk of severe/critical COVID-19 that is 1.195 times greater than that of females.
(iv) Page 10: Is “rapid antigen testing” –even under favorable conditions – a reliable test at all?
We added: “While less accurate than nucleic acid tests, rapid antigen tests appear to be best suited for identifying SARS-CoV-2 infection in individuals who are symptomatic in high prevalence settings [8,9]. Perhaps we could have enrolled additional participants during the peak of the Delta or Omicron waves if we had used antigen testing to screen symptomatic adolescents and youth.”
(v) Line 293 , …. rare, instead of rate.
Thank you for catching this. We corrected this.
References
- Madewell, Z.J.; Yang, Y.; Longini, I.M., Jr; Halloran, M.E.; Dean, N.E. Factors Associated With Household Transmission of SARS-CoV-2: An Updated Systematic Review and Meta-analysis. JAMA Network Open 2021, 4, e2122240-e2122240, doi:10.1001/jamanetworkopen.2021.22240.
- Avanzato, V.A.; Matson, M.J.; Seifert, S.N.; Pryce, R.; Williamson, B.N.; Anzick, S.L.; Barbian, K.; Judson, S.D.; Fischer, E.R.; Martens, C.; et al. Case Study: Prolonged Infectious SARS-CoV-2 Shedding from an Asymptomatic Immunocompromised Individual with Cancer. Cell 2020, 183, 1901-1912.e1909, doi:https://doi.org/10.1016/j.cell.2020.10.049.
- Khatamzas, E.; Antwerpen, M.H.; Rehn, A.; Graf, A.; Hellmuth, J.C.; Hollaus, A.; Mohr, A.W.; Gaitzsch, E.; Weiglein, T.; Georgi, E.; et al. Accumulation of mutations in antibody and CD8 T cell epitopes in a B cell depleted lymphoma patient with chronic SARS-CoV-2 infection. Nature Communications 2022, 13, doi:10.1038/s41467-022-32772-5.
- Maiese, A.; Frati, P.; Del Duca, F.; Santoro, P.; Manetti, A.C.; La Russa, R.; Di Paolo, M.; Turillazzi, E.; Fineschi, V. Myocardial Pathology in COVID-19-Associated Cardiac Injury: A Systematic Review. Diagnostics 2021, 11, 1647.
- Maiese, A.; Manetti, A.C.; Bosetti, C.; Del Duca, F.; La Russa, R.; Frati, P.; Di Paolo, M.; Turillazzi, E.; Fineschi, V. SARS-CoV-2 and the brain: A review of the current knowledge on neuropathology in COVID-19. Brain Pathology 2021, 31, e13013, doi:https://doi.org/10.1111/bpa.13013.
- Maiese, A.; Manetti, A.C.; La Russa, R.; Di Paolo, M.; Turillazzi, E.; Frati, P.; Fineschi, V. Autopsy findings in COVID-19-related deaths: a literature review. Forensic Science, Medicine and Pathology 2021, 17, 279-296, doi:10.1007/s12024-020-00310-8.
- Czerkies, M.; Kochańczyk, M.; Korwek, Z.; Prus, W.; Lipniacki, T. Respiratory Syncytial Virus Protects Bystander Cells against Influenza A Virus Infection by Triggering Secretion of Type I and Type III Interferons. Journal of Virology 2022, 96, e01341-01322, doi:doi:10.1128/jvi.01341-22.
- Ye, Q.; Shao, W.X.; Meng, H.Y. Performance and application evaluation of SARS-CoV-2 antigen assay. Journal of Medical Virology 2022, 94, 3548-3553, doi:10.1002/jmv.27798.
- Peeling, R.W.; Heymann, D.L.; Teo, Y.Y.; Garcia, P.J. Diagnostics for COVID-19: moving from pandemic response to control. Lancet 2022, 399, 757-768, doi:10.1016/s0140-6736(21)02346-1.